# Non-Traditional Pathways for Platelet Pathophysiology in Diabetes: Implications for Future Therapeutic Targets

**DOI:** 10.3390/ijms23094973

**Published:** 2022-04-29

**Authors:** Rebecca C. Sagar, Ramzi A. Ajjan, Khalid M. Naseem

**Affiliations:** Leeds Institute of Cardiovascular and Metabolic Medicine, Faculty of Medicine and Health, University of Leeds, Leeds LS2 9JT, UK; r.ajjan@leeds.ac.uk (R.A.A.); k.naseem@leeds.ac.uk (K.M.N.)

**Keywords:** platelets, diabetes, thrombo-inflammation

## Abstract

Cardiovascular complications remain the leading cause of morbidity and mortality in individuals with diabetes, driven by interlinked metabolic, inflammatory, and thrombotic changes. Hyperglycaemia, insulin resistance/deficiency, dyslipidaemia, and associated oxidative stress have been linked to abnormal platelet function leading to hyperactivity, and thus increasing vascular thrombotic risk. However, emerging evidence suggests platelets also contribute to low-grade inflammation and additionally possess the ability to interact with circulating immune cells, further driving vascular thrombo-inflammatory pathways. This narrative review highlights the role of platelets in inflammatory and immune processes beyond typical thrombotic effects and the impact these mechanisms have on cardiovascular disease in diabetes. We discuss pathways for platelet-induced inflammation and how platelet reprogramming in diabetes contributes to the high cardiovascular risk that characterises this population. Fully understanding the mechanistic pathways for platelet-induced vascular pathology will allow for the development of more effective management strategies that deal with the causes rather than the consequences of platelet function abnormalities in diabetes.

## 1. Introduction

Cardiovascular complications represent the leading cause of morbidity and mortality in patients with diabetes (DM), increasing the economic burden on healthcare systems [1,2,3]. There is an elevated risk of a first vascular event in individuals with diabetes, and outcomes following vascular ischaemia are inferior compared to those with normal glucose regulation [4]. The increased cardiovascular morbidity in subjects with DM is associated with profound metabolic and functional changes in the cells of the vasculature. In the context of the current evidence, the premature and more extensive vascular disease is coupled with a prothrombotic environment in which platelet hyperactivity is thought to play a key role in the suboptimal clinical outcomes. Oxidative stress, dyslipidaemia, and a combination of insulin resistance/hyperglycaemia, typical of patients with DM, have been proposed to contribute to abnormal platelet function [5,6]. The pervading view of the role of platelets in the development of cardiovascular complications in this cohort is focused on their contribution to arterial thrombosis at sites of plaque rupture. Antiplatelet agents, including aspirin, ticagrelor, and clopidogrel, are routinely used to suppress platelet function and reduce the risk of atherothrombosis [7,8,9]. However, there is emerging evidence to indicate that platelets may also contribute to the pervasive low-grade inflammation that promotes increased cardiovascular risk in DM [10,11,12,13].

Platelets possess a full repertoire of inflammatory functions and a diverse array of mechanisms for the transcellular transfer of inflammatory factors, allowing them to coordinate the interactions of endothelial cells with circulating immune cells [14]. The release of platelet α-granules results in the surface expression of P-selectin and the release of preformed chemokines such as CCL3, CCL5, CCL5, platelet factor 4(PF4), PAF, and CXCL10, amongst others [14]. P-selectin facilitates heterotypic interactions with both endothelial cells and leukocytes through P-selectin glycoprotein ligand-1. These bioactive mediators trigger the expression of proinflammatory gene products in both endothelium and leukocytes [14]. Through the release of vasoactive factors and the formation of heterotypic cell complexes, platelets act as a focal point for vascular inflammation, enabling the recruitment of leukocytes to the endothelium and their transmigration to the subendothelial space [15]. However, the precise mechanisms of platelet-driven inflammation in individuals with DM are unclear.

The role that platelets play in these various pathways has led to the emerging evidence around their involvement in thrombo-inflammation. This concept is distinct from inflammation alone in that it refers to pathological states, when, following vascular injury there is a coordinated response from both thrombotic and inflammatory pathways to ensure the pathological process remains limited to the site of injury, allowing for effective and complication-free healing [16].

This narrative review highlights the role of platelets in the inflammatory and immune responses that contribute to cardiovascular disease in diabetes. In particular, this work discusses the potential effects of DM on platelet-driven inflammation, the principles of platelet reprogramming in diabetes, and the potential therapeutic targets that these pathways may provide.

## 2. Pathophysiology of Vascular Disease in Diabetes

The fundamentals of the pathophysiology behind inflammation-driven vascular damage in diabetes are key to identifying potential pathways for therapeutic targets to prevent and treat vascular complications in diabetes. Endothelial dysfunction is a key abnormality in diabetes and contributes to both a proinflammatory and prothrombotic environment that promotes vascular occlusive disease [5,15,17]. A close association between endothelial dysfunction and platelet activity has been repeatedly demonstrated [17,18], and recent evidence suggests this relationship is bidirectional.

### 2.1. Endothelial Dysfunction and Atheroma Formation

The endothelium is a principal regulator of a number of thrombotic and non-thrombotic pathways [19,20]. Collectively, the endothelia act as a bioactive organ that controls the function of blood cells, the integrity of the vascular wall, and vascular reactivity. Critical to these functions are the vasoactive mediators, nitric oxide (NO) and prostacyclin [21,22,23,24]. The tonic release of these mediators prevents vascular inflammation by ensuring platelet quiescence and preventing platelet-mediated immune cell infiltration of the subendothelial space, factors that are critical to preventing vascular inflammation. A key characteristic of endothelial dysfunction is the lack of bioavailable NO and PGI_2_, leading to the loss of their athero-protective effects. When inflamed, endothelial cells increase the cell surface expression of cell adhesion molecules and release chemotactic messengers that promote the recruitment and reaction of monocytes into the subendothelial space and their subsequent transformation into macrophages [18,25,26]. Endothelial dysfunction occurs as a result of several metabolic features typical of diabetes, including hyperglycaemia, insulin resistance, and the resulting increased oxidative stress [5,23]. There is also an increase in permeability, which potentially allows for an increased accumulation of low-density lipoproteins (LDLs) in the vessel wall, where they are retained and prone to oxidative attack. The subsequent unregulated uptake by macrophages of oxidised-LDL results in the formation of foam cells. These cells secrete cytokines, including interleukin-6 (IL-6) and tumour necrosis factor (TNF)-α [25,26], further enhancing the proinflammatory environment [27]. As this process continues, atherosclerotic plaques continue to grow and eventually rupture, causing the activation of platelets; this drives clot formation.

### 2.2. Intravascular Thrombus Formation

Upon the rupture of the atherosclerotic plaque, a cascade of events ensues that results in the activation of both the cellular and acellular arms of coagulation, promoting thrombus formation. NO once again plays a vital role in the regulation of platelet adhesion and aggregation, normally preventing thrombus formation by inhibiting platelet adhesion and aggregation, while also promoting the disaggregation of pre-formed platelet aggregates [28]. Thus, when NO bioavailability falls in diabetes, the consequence is an increased potential for platelet activation and thrombus formation, also contributing to an inflammatory state [25]. The activation of platelets facilitates the localised activation of the coagulation cascade and the generation of a fibrin network that stabilises the thrombus. DM is characterised by dense fibrin networks and hypofibrinolysis [29,30,31], which contribute to vascular complications and adverse clinical outcomes in this population [32,33] (Figure 1).

## 3. Diabetes-Related Mechanistic Pathways Modulating Thrombo-Inflammatory Function of Platelet

In patients with DM, particularly T2DM, a number of changes in the receptor and signal transduction function have been described that contribute to platelet dysfunction. We discuss below the main pathways that are likely to operate in diabetes and which are responsible for modulating platelet function, with a focus on thrombo-inflammatory pathways.

### 3.1. Insulin and the Insulin Receptor

The majority of patients with diabetes have T2DM, typically characterised by insulin resistance and consequent hyperinsulinaemia [6,17,34]. These features may have often been present for decades prior to a formal diagnosis of T2DM [35]. Platelets express the insulin receptor on their surface, although the exact function of the receptor is yet to be fully determined [5,23]. In healthy non-overweight people, insulin binding to its receptor results in the inhibition of platelet activation, secondary to the intracellular translocation of magnesium [35]. This pathway is mediated by the activation of insulin receptor substrate (IRS-1) via tyrosine phosphorylation, which in turn increases cytosolic cyclic adenosine monophosphate (cAMP), a key platelet inhibitor. The increased cytosolic cAMP concentration is proposed to reduce activation signalling by the ADP receptor P2Y_12_, thereby suppressing platelet activity. Impaired insulin signalling as a result of insulin resistance (IR), seen in individuals with T2DM, or absolute insulin deficiency occurring in T1DM, leads to disinhibited platelet activation [23,36]. While studies on platelet reactivity in T1D are both limited and conflicting [37,38,39,40], the lower plasma level of magnesium in these individuals may contribute to altered platelet function [41]. Alterations in insulin receptor signalling in insulin resistance can also reduce cAMP levels, which results in increased cytosolic calcium concentration, resulting in platelet hyperreactivity [42].

### 3.2. Nitric Oxide and Reactive Oxygen Species

Hyperglycaemia and insulin resistance, as well as dyslipidaemia and obesity, commonly seen in patients with DM, also drive cardiovascular disease through vascular inflammation. These factors result in an imbalance between the production of endothelial NO synthase (eNOS), derived NO, and the elevated production of reactive oxygen species (ROS), leading to the disruption of this vital homeostatic environment [22,43]. The increased accumulation of ROS results in the inactivation of NO to form peroxynitrite, following the generation of superoxide anion. This key event, driven by both insulin resistance and hyperglycaemia, leads to a reduction in NO bioavailability, which is further exacerbated by peroxynitrite driving the uncoupling of eNOS, with a preferential production of ROS [15]. Peroxynitrite has been shown to result in the damage and death of both endothelial and vascular smooth muscle cells, and thus, has been linked to the development of cardiovascular complications in diabetes [44,45,46]. Another mechanism contributing to reduced endothelium-derived NO in diabetes is the decreased activity of eNOS [19,23], as a result of both excess ROS production and increased protein kinase C (PKC) activity. Given the vasculo-protective actions of NO, a reduction in its bioavailability is associated with adverse cardiovascular outcomes [23,47]. Reduced NO levels coupled with elevated ROS levels promote the production of transcription nuclear factor kappa B (NF-kB), a transcription factor involved in several cellular pathways in endothelial cells, resulting in the increased production of chemokines and cytokines that are potentially associated with inflammation [19,23]. The increased expression of NF-kB has been shown to enhance the expression of leukocyte adhesion molecules in endothelial cells while also stimulating the production of chemokines and cytokines, further contributing to an inflammatory state and atherosclerotic changes [48]. The decreased bioavailability of NO in DM could also potentially lead to a loss of platelet activation pathways. In diabetic mice, the inhibition of NO synthase led to increased fibrinogen-platelet binding and the expression of activation markers CD40-L and P-selectin [49]. Improving endothelial NO availability resolved these observed pathological changes. Indeed, some studies, but not all, have demonstrated reduced NO in patients with DM [50,51]. This further supports the impact of both NO bioavailability on platelet hyperreactivity as well as the impact of diabetes on NO production.

In addition to reduced levels of NO, the accumulation of ROS leads to the activation of other additional pathways that contribute to inflammation [15,52], particularly the generation of advanced glycation end products (AGEs) [52,53]. The production of AGEs affects protein function and also activates the receptor for AGEs (RAGEs). AGEs further drive ROS production, and RAGE activation leads to increased superoxide anion production, both of which additionally contribute to diminished NO. PKC activation has been linked to hyperglycaemia and leads to changes that contribute to vascular disease, including inflammation and platelet hyperreactivity, as well as alterations in angiogenesis, cell growth, and apoptosis [54]. Elevated PKC activity has been demonstrated in the platelets of healthy controls left in hyperglycaemic conditions, although this has been variable in patients with T2D [55]. PKC activation drives ROS generation via NADPH oxidase-mediated superoxide production [56]. It also decreases eNOS activity, with the resultant diminished NO production described above. Along with reduced vasodilation through these mechanisms, PKC also drives the elevated production of the vasoconstrictor, endothelin-1, which promotes vasoconstriction and platelet aggregation [54].

### 3.3. Platelet Activation and P-Selectin

It has been well-established that individuals with both T1D and T2D display enhanced platelet activation compared to platelets taken from healthy individuals. Much early evidence has come from studies focussing on thromboxane (TXA) biosynthesis [57,58]. Davi et al. crucially demonstrated that DM, amongst other risk factors for CVD, such as hypertension, causes a persistent state of platelet activation, measured through thromboxane biosynthesis. This may, in turn, also suggest a persistent secretion of inflammatory mediators [59]. Further to this, other studies have also demonstrated enhanced TXA synthesis in the context of post-prandial hyperglycaemia alone [60].

More recently, many studies have used P-selectin as a marker of platelet activation. The activation of platelets upregulates P-selectin expression on cell membranes. The binding of P-selectin to P-selectin glycoprotein ligand-1 on leukocytes is the primary pathway in the formation of heterotypic platelet–leukocyte aggregates (specifically, the monocyte and neutrophil subtypes). Platelet–monocyte aggregates have been the most widely studied, largely due to the fact they are the most stable platelet–leukocyte aggregates. These aggregates have been shown to further enhance platelet adhesion, thereby contributing to the prothrombotic environment through excess platelet aggregation and interaction with the endothelium [14]. P-selectin-mediated platelet–leukocyte interaction also activates inflammatory processes, upregulating the gene expression of proinflammatory cytokines and integrins that contribute to vascular damage [61].

Individuals with T1D have been shown to have higher circulating levels of both P-selectin and platelet–monocyte aggregates compared to healthy controls, without an increase in platelet–neutrophil aggregates [13]. Medium-term hyperglycaemia, measured through glycated haemoglobin (HbA1c), correlates with P-selectin expression and platelet–monocyte aggregate formation, directly implicating raised glucose levels in platelet-mediated inflammation. A further study demonstrated that experimental hyperinsulinaemia and hyperglycaemia in healthy patients are associated with increased platelet–monocyte, but not platelet–neutrophil aggregates, suggesting that both insulin resistance and hyperglycaemia affect the proinflammatory properties of platelets [62]. To further emphasise the importance of hyperglycaemia, platelet reactivity has been shown to decrease (measured by reduced P-selectin expression) as a result of improvements in glycaemic control [63]. Studies have also shown elevated P-selectin levels in patients with T2DM, with Eibl et al. demonstrating a significant reduction of P-selectin levels following improvement in glycaemic control (assessed as HbA1c) after 3 months [64,65].

### 3.4. CD40-Ligand

CD40L, a tumour-necrosis factor ligand, is stored in platelets and is rapidly expressed on the platelet surface before cleavage [66]. CD40-L interacts with cells displaying the CD40 receptor, which includes a number of important inflammatory cells, such as monocytes and macrophages. The binding of CD40 to its ligand is potentially very important since it induces a signalling response that drives the synthesis and release of a number of key chemokines and cytokines from inflammatory cells, including IL-6 and IL-8 [67]. It was observed that both platelet CD40L expression and platelet–monocyte aggregates are elevated in patients with T1D compared with healthy controls [68]. Consistent with this observation, elevated circulating CD40L in patients with DM (both T1D and T2D) compared to healthy age-matched healthy controls was also observed [69]. There is further evidence to suggest that this is another potential pathway by which inflammation is increased in DM, with healthy participants demonstrating an increased number of CD40L on platelets following the induction of a hyperglycaemic and hyperinsulinaemic environment [62]. Enhanced platelet activation in obese individuals with normal blood glucose levels emphasises the importance of insulin resistance in modulating platelet function. The evidence of increased platelet activity has been shown in obese individuals with elevated levels of plasma CD40-ligand (CD40L), higher urinary thromboxane metabolite, as well as higher levels of platelet-derived microparticles, and these elevated markers have been shown to improve with weight loss and better glycaemic control [70,71,72,73].

### 3.5. Toll-like Receptors and Immune Response

The relatively recent identification of the expression of Toll-like receptors (TLR) in human and mouse platelets supports the theory that platelets possess immune-related capabilities beyond haemostasis. These receptors, which recognise a plethora of endogenous damage-associated molecular patterns (DAMPs) and exogenous pathogen-associated molecular patterns (PAMPs), allow platelets to play a prominent role in the immune surveillance of the vasculature. Their enhanced expression on platelets has now been repeatedly demonstrated at both the mRNA and protein level in a number of disease states, including infection (bacterial and viral), as well as in CVD [63,74,75]. TLR expression drives the activation of platelets and induces aggregation in addition to the release of inflammatory cytokines and the activation of the NF-kB pathway [76,77]. Particularly relevant to CVD, studies have demonstrated elevated platelet TLR-2 mRNA expression and protein production in patients with acute coronary syndrome [74,75], linking TLRs not only to chronic but also acute vascular pathology. The mechanism by which TLRs potentially contribute to platelet inflammatory function is beginning to emerge and may be related to an increased synthetic capacity. In immune cells, TLR activation is linked to the activation of inflammasomes, particularly the NOD-like receptor protein 3 (NLRP3) inflammasome, which generates interleukin 1β (IL-1β) [78]. Metabolic DAMPs, such as AGEs, palmitate, and glucose, often elevated in T2DM, typically drive NLRP3 activation, and thus, IL-1β synthesis [79]. The activation of the NLRP3 inflammasome has been shown in monocytes from patients with T2DM, leading to increased IL-1β [80]. Interestingly, this was modulated by treatment with metformin. A number of studies have demonstrated that metabolic dysregulation, such as obesity, leads to the activation of the NLRP3 inflammasome in various cells, including PBMCs and endothelial cells. It has been postulated that the metabolic environment of T2D, characterised by hyperglycaemia and hyperinsulinaemia, is a key activator of the NLRP3 inflammasome, particularly given its upregulation in this population [81]. One study demonstrated that NLRP3 activation was increased in monocyte-derived macrophages from patients with diabetes [81] as well as in the endothelial cells of diabetic mice [82]. Further to this, the NLRP3 knockdown in a mouse model for diabetic atherosclerosis was shown to have reduced endothelial inflammation and lower atherosclerotic lesion burden [82]. Additionally, NLRP3 inflammasome activation is enhanced in patients with newly diagnosed diabetes compared to healthy matched controls. The same study also showed that improvement in the glycaemic control in this patient cohort led to significant reductions in NLRP3 inflammasome activity [82].

Elevated levels of circulating free fatty acids, often seen in diabetes, can bind to TLRs, inducing an increased expression of key inflammatory molecules, including IL-6 and TNF-α, as a result of the activation of the described NF-kB pathway [15,54,83], both of which are known to result in abnormal platelet function [84].

The various pathways modulating the thrombo-inflammatory function of platelets are summarised in Figure 2.

## 4. Platelet Metabolic Reprogramming in Diabetes: Future Therapeutic Targets?

### 4.1. Metabolic Reprogramming and Platelet Bioenergetics

The links between metabolism and inflammation have been shown, predominantly in immune cells. Immunometabolism is a term relating to the interplay between metabolic regulation and immune function [85]. Evidence has shown that in immune cells, a switch can occur in metabolic pathways from oxidative phosphorylation to aerobic glycolysis, and this may drive a persistent inflammatory state [86]. The abundance of nutrients, with hyperglycaemia and elevated circulating free fatty acids seen in DM, have been proposed as potential drivers of this ‘immunometabolic reprogramming’, resulting in sustained low-grade inflammation [86]. Given the growing evidence implicating platelets in immune responses, it can be hypothesised that similar changes occur in these two cell types in response to pathological changes [87,88].

Platelet activation, in response to both thrombotic and inflammatory processes, is energetically expensive, and thus, requires a significantly enhanced generation of ATP via glycolysis and oxidative phosphorylation. Specific disease states have been shown to increase platelet glycolysis and oxidative phosphorylation, evidenced by an elevated extracellular acidification rate (ECAR) and increased oxygen consumption rate (OCR), respectively [89,90]. Glucose is a key and potent energy source driving these processes, and therefore, hyperglycaemia in DM may drive these processes, whilst improved glycaemic control can reverse these changes, at least partly [91]. Although little evidence exists to demonstrate changes in the bioenergetics of platelets in patients with DM, a study investigated these changes in the platelets of patients with sickle cell disease [92]. The results suggested that there is variation in the bioenergetic programming amongst individuals and that there is metabolic adaptability within platelets to meet energy demands that are particularly affected in disease states. Of particular note was the observation of a dysfunctional relationship between this metabolic ability to meet energy demands in those with sickle cell disease compared to healthy controls, demonstrated by a loss of the relationship between basal OCR and ATP-linked OCR and suggesting a reduction in the maximal respiration capacity despite demand [92]. It is possible that other disease states, including DM, may see a similar pattern.

### 4.2. Altered Platelet mRNA and Protein Expression

In addition to platelet metabolism, platelet transcriptomics and proteomics have been growing areas of interest [93,94] and may prove to have a role in tailored therapies in individuals at risk of CVD. It has been well-established that platelets, whilst anucleate, still have mRNA, which, once spliced into mature RNA, can be translated into proteins. Given the complex conditions within the inflammatory and metabolic milieu of the blood of patients with DM, it is possible that platelets can respond by altering their proteome.

Alterations in mRNA expression and subsequent protein transcription have been linked to a number of disease states and may help to establish whether and how the disease environment specific to DM can result in ‘immunometabolic reprogramming’ [94]. Such studies have been undertaken in patients with sickle cell disease and systemic lupus erythematosus (SLE), demonstrating differences in protein expression compared to healthy volunteers, directly affecting platelet function [95,96]. Similarly, platelets from patients with obesity and HIV have been shown to have an altered platelet transcriptome and proteome [97,98,99]. In the case of HIV, the enhanced platelet expression of ABCC4 is directly associated with platelet hyperactivity [97]. Early studies in those with ACS were shown to have elevated platelet TRP14 and CD69, which was also associated with hyperactivity [100]. The reverse engineering of these studies demonstrated that TRP14 is a ligand for platelet CD36 and drives thrombosis in hyperlipidaemic mice. It is yet unclear if similar changes are associated with platelets from people with DM. However, platelet mRNA may represent a useful tool for both the prognostication and/or diagnosis of vascular risk as a result of functional platelet changes in certain patient groups.

### 4.3. Platelet-Specific miRNA

Several studies have also investigated miRNAs and their role in endothelial dysfunction in diabetes. Platelet miR-223 has been implicated in the ADP-receptor P2Y_12_ pathway [101], where reduced levels in patients with T2D compared to healthy controls are associated with increased activity of the receptor and enhanced platelet reactivity [102]. miR-26b and miR-140 are believed to target P-selectin mRNA, driving excess P-selectin levels and, thus, heightening platelet activity [103].

Platelet miR-223 has been shown to be reduced in patients with DM as well as in mouse models of DM. miR-223 knockout mice were shown to have increased platelet aggregation and thrombus formation compared to wild-type mice [104,105]. However, Parker et al. investigated patients with T2DM receiving antiplatelet therapy (aspirin, clopidogrel, prasugrel) and found reduced levels of miR-223, miR-197, miR-24, and miR-191 in those receiving prasugrel compared to aspirin, a treatment that was associated with more profound platelet suppression. Furthermore, in those patients on aspirin or prasugrel with a history of CVD, there were lower levels of miR-197 compared to individuals without a CVD history, which may be of use as a potential biomarker in this cohort [106]. Another study examined miRNA in patients with DM with and without ischaemic stroke. In those who had an ischaemic stroke and DM or DM alone, there were lower circulating levels of both platelet miR-223 and miR-146a, which was associated with increased platelet activation compared to those patients with only an ischaemic stroke or healthy controls. The conditions of hyperglycaemia have also been shown to downregulate all three miRNAs, miR-223, miR-26b, and miR-140. The reduced levels of these miRNAs lead to the upregulated expression of the various prothrombotic receptors in platelets, including P2Y_12_ and P-selectin [102,103], and have been linked to elevated platelet activation measured through surface P-selectin expression.

In addition to representing potential biomarkers, the affected pathways driving platelet reactivity may be useful in developing therapeutic targets to reduce platelet-driving thrombo-inflammation [102,103]. Therefore, miRNA may be used as a marker of vascular disease or, alternatively, to monitor the response to specific therapies. This may, in turn, lead to therapies that enhance or suppress specific miRNA as a new management strategy to reduce vascular risk.

### 4.4. Mitochondrial Dysfunction

As previously described, oxidative stress is a key aspect of the cellular environment of patients with DM. This coupled with the driving force of hyperglycaemia and disordered insulin production/function, altering platelet reactivity and the inflammatory profile, also contributes to mitochondrial dysfunction [107,108,109]. Increased oxidative stress has been demonstrated in T2DM [110], affecting platelet mitochondria, which in turn, increases ROS production, creating a vicious cycle [111]. Lee et al. demonstrated that elevated oxidative stress increased the protein phosphorylation of p53 in pooled platelets from patients with DM. The increased phosphorylated p53 and translocation to mitochondria is a driver of mitochondrial dysfunction in the platelets of patients with DM as well as elevated platelet apoptosis [108,112]. This increase in the phosphorylation of p53 in DM platelets has also been shown to be mediated by aldose reductase in both human and mouse models and also contributes to platelet activation in DM [112,113]. Further to this, the blocking of aldose reductase has been shown to reduce thromboxane release in response to collagen and, thus, reduces platelet activation, demonstrating its potential key role in driving not only mitochondrial dysfunction in platelets but also the levels of activation [114].

Given the importance of oxidative stress in the pathways responsible for vascular pathology, several studies have investigated the role of antioxidants with variable and inconclusive results. Limited data suggest an association between increasing dietary antioxidant nutrients and protection against cardiovascular disease [115]. Specifically, in patients with diabetes, low carotenoid intake has been linked to reduced insulin resistance [116]. In contrast, the HOPE trial failed to show any benefit of Vitamin E on cardiovascular outcomes or mortality in high-risk individuals with diabetes [117]. The exact reasons for the lack of positive outcomes with the use of antioxidants in these trials are not fully clear. It may be related to studying highly heterogeneous populations, with antioxidants having variable and inconsistent effects. It is also possible that different doses of antioxidants are required according to various factors, including DM duration, glycaemic control, and therapies, as well as the presence of vascular complications, which have never been explored.

Further to this mitochondrial dysfunction, the maladaptive changes in the metabolism seen in DM as well as other disease states, such as obesity, with readily available fatty acids [79], have recently been linked to the activation of the aforementioned NLRP3 inflammasome and may link nutrient excess to inflammation and inflammatory pathways. Therefore, this previously described immunometabolic reprogramming may be a potential explanation for the upregulation of the NLRP3 inflammasome seen in DM [81]. Recent data also support the fact that elevated ROS, as a result of mitochondria, drive NLRP3 inflammasome activation. Lee et al. demonstrated that monocyte-derived macrophages in patients with T2DM have much higher mRNA and protein expression of NLRP3 and IL-1β compared to healthy controls. Following 2 months of metformin treatment with associated HbA1c and fasting glucose improvements, the levels of IL-1β maturation and production following stimulation fell [81]. Similarly, platelets from subjects with IR and obesity were found to have an upregulated expression of mRNA for IL-1β and NLRP3 inflammasome [118].

The relative importance of the role of platelet function in the vascular risk in patients with DM is all the more heightened by the successful use of antiplatelet treatment, particularly in secondary prevention. Thus, dysfunction in platelet activity not only drives the vascular risk itself but may have implications for the efficacy of these treatment options, as seen by the apparent aspirin resistance in this patient cohort [119,120]. Having a fundamental understanding of the translational changes affecting platelet function may also help to mitigate these potentially negative clinical outcomes.

## 5. Conclusions

While modern management strategies have reduced cardiovascular complications in patients with diabetes, long-term outcomes remain inferior compared to individuals with normal glucose metabolism. Platelets play a key role in contributing to pathological vascular occlusion in diabetes, and it is now clear that platelet function stems far beyond the traditional role in haemostasis, with important effects not only on thrombosis but also on both the immune and inflammatory processes.

While some studies have shown reduced platelet activation by improving glycaemic control, this appears to be partial, with the added complication that aggressive glycaemic control induces hypoglycaemia, which is itself both prothrombotic and proinflammatory. A number of methods have been used to test the thrombotic properties of platelets, reviewed elsewhere [8], but tests to measure the inflammatory characteristics of these cells remain an area for future work.

This review highlights a number of platelet-specific pathways that operate in diabetes and drive the thrombo-inflammatory milieu. In particular, platelet reprogramming in diabetes transforms these cells to display not only prothrombotic but also proinflammatory characteristics. This in turn contributes to the ongoing vascular pathology and results in premature and more severe vascular disease in this population. Rather than dealing with the consequences of platelet reprogramming in diabetes, which can be associated with unwanted side effects, a more efficient strategy is to understand the pathways leading to these changes. This in turn will allow for effective risk stratification and the development of targeted therapies. For example, the identification of potentially important platelet miRNA/mRNA may help in risk stratification and the intensification of treatment, accordingly. Targeting mitochondrial dysfunction offers another novel management strategy that has the potential to normalise platelet function and limit vascular pathology. Developing therapies that target individual-specific pathological processes will help to safely and effectively reduce the thrombo-inflammatory milieu in diabetes and improve outcomes in this high-risk population.

## Figures and Tables

**Figure 1 ijms-23-04973-f001:**
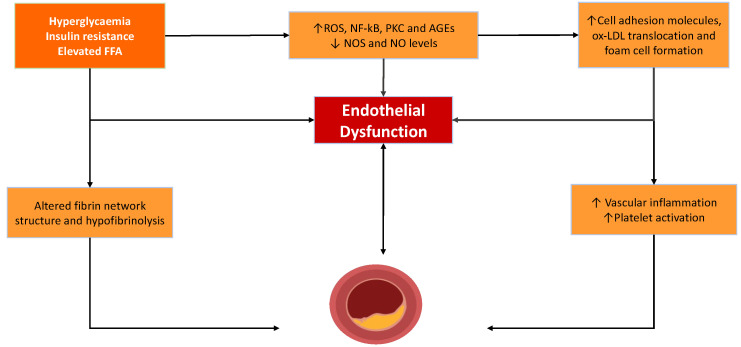
Factors contributing to endothelial dysunfction, elevated vascular inflammation, platelet activation driving the development of atheroma and intravascular thrombus. Abbreviations; FFA: Free Fatty Acids, ROS: Reactive Oxygen Species, NF-kB: nuclear factor kappa-light-chain-enhacer of activated B cells, PKC: protein kinase C, AGEs: Advanced Glycation End Products, ox-LDL: oxidised-low density lipoprotein.

**Figure 2 ijms-23-04973-f002:**
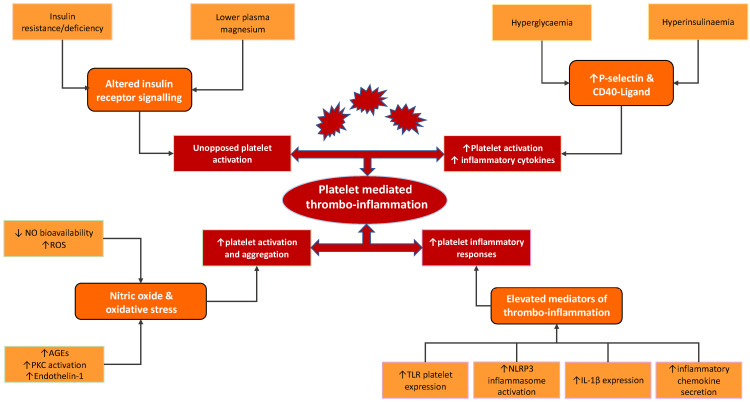
Factors present in diabetes that drive unopposed platelet activation, increased platelet activation and release of inflammatory cytokines, increased platelet aggregation and also elevated platelet inflammatory responses. Abbreviations; NO: nitric oxide, ROS: reactive oxygen species, AGEs: Advanced Glycation End Products, PKC: protein kinase C, TLR: Toll-like receptor, NLRP3: NOD-like receptor protein 3, IL-1β: interleukin-1β.

## Data Availability

Not applicable.

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
