# Peer review of "Non-Traditional Pathways for Platelet Pathophysiology in Diabetes: Implications for Future Therapeutic Targets"

_ijms, 2022, doi:10.3390/ijms23094973_

Round 1

Reviewer 1 Report

This review is timely in the light of increasing evidence for platelet role in thrombo-inflammatory states. The authors have done well to put this work together. I have some recommendations to further improve the article.

Major comment

The article mentions severally the mechanistic significance of reduced nitric oxide (NO) bioavailability. They also highlighted the overproduction of superoxide in a disease marked by oxidative stress. While the authors rightly mentioned that NO, in the presence of superoxide will lead to peroxynitrite (ONOO-) formation, there is no mention of the cellular actions of peroxynitrite in the review.  The formation of ONOO- from NO and superoxide is perhaps the fastest reaction known in biology and the formed ONOO- is both a potent oxidising and nitrating agent, with potent cellular actions1-3. Clearly, peroxynitrite is likely involved in the pathophysiology of diabetes. Can the authors briefly summarise what is known, and how the cellular actions of ONOO- relate to their proposed model of thrombo-inflammation in diabetes? This will fit well with section 3.2 or perhaps as a separate section 3.3.

Minor comment

  1. At some point, maybe in the Conclusion section, the authors should make clear their position on what is driving what? Is the disease or inflammation driving platelet function or vice versa, or something else? the authors have already provided the evidence.
  2. At lines 53-57, are the authors referring to normal haemostasis? Which is a normal /physiological response involved in complication-free healing.
  3. Several sentences are hard to follow, please check and read through manuscript and simplify to allow for easy reading.
  4. Check that all is ok with lines 59-60, 81, 173, 311
  5. Is the word ‘both’ redundant in line 73?
  6. Line 103.. did the authors mean thrombosis instead of haemostatic plug? Haemostatic plugs are formed mainly in the extravascular space while thrombosis is predominantly intravascular.
  7. Lines 195-197; authors should explain in one or 2 lines how adhesion contributes to prothrombotic environment.
  8. In concluding the review, what platelet mediators will the authors suggest targeting to control platelets’ contribution to thrombo-inflammation in DM

References

  1. Agbani EO, Coats P, Mills A, Wadsworth RM. Peroxynitrite stimulates pulmonary artery endothelial and smooth muscle cell proliferation: Involvement of ERK and PKC. Pulmonary Pharmacology & Therapeutics. 2011;24(1):100-109.
  2. Agbani EO, Coats P, Wadsworth RM. Acute Hypoxia Stimulates Intracellular Peroxynitrite Formation Associated With Pulmonary Artery Smooth Muscle Cell Proliferation. Journal of Cardiovascular Pharmacology. 2011;57(5):584-588 510.1097/FJC.1090b1013e3182135e3182131b.
  3. Agbani E, Coats P, Wadsworth RM. Threshold of peroxynitrite cytotoxicity in bovine pulmonary artery endothelial and smooth muscle cells. Toxicology in Vitro. 2011;25(8):1680-1686.

Author Response

The authors would like to thank reviewer 1 for their positive reception of our review and their helpful comments which have been addressed as follows:

  1. The article mentions severally the mechanistic significance of reduced nitric oxide (NO) bioavailability. They also highlighted the overproduction of superoxide in a disease marked by oxidative stress. While the authors rightly mentioned that NO, in the presence of superoxide will lead to peroxynitrite (ONOO-) formation, there is no mention of the cellular actions of peroxynitrite in the review.  The formation of ONOO- from NO and superoxide is perhaps the fastest reaction known in biology and the formed ONOO- is both a potent oxidising and nitrating agent, with potent cellular actions1-3. Clearly, peroxynitrite is likely involved in the pathophysiology of diabetes. Can the authors briefly summarise what is known, and how the cellular actions of ONOO- relate to their proposed model of thrombo-inflammation in diabetes? This will fit well with section 2or perhaps as a separate section 3.3.

Response: As the subject of this review is quite extensive, we have tried to keep things focussed and thus it was difficult to go into detail in some areas which could have been greatly expanded. We thank the reviewer for these helpful suggestions and have added a few sentences in section 3.2, to include reference to the mentioned areas and have included the proposed references.

  1. At some point, maybe in the Conclusion section, the authors should make clear their position on what is driving what? Is the disease or inflammation driving platelet function or vice versa, or something else? the authors have already provided the evidence.

Response: We have amended the wording of the conclusion to reflect this suggestion and we would like to thank the reviewer for highlighting this.

  1. At lines 53-57, are the authors referring to normal haemostasis? Which is a normal /physiological response involved in complication-free healing.

Response: This refers to the role of platelets in the pathological thrombotic effects seen in cardiovascular complications in diabetes, rather than the focus of this review on the wider role of platelets in both inflammation and immune pathways also contributing to these cardiovascular complications. We have added amended to add clarity to this and thank the reviewer for highlighting this.

  1. Several sentences are hard to follow, please check and read through manuscript and simplify to allow for easy reading

Response:  We have edited the review to shorten/simplify longer sentences and hope we added the required clarity.

  1. Check that all is ok with lines 59-60, 81, 173, 311

Response: Amendments have been made to these lines where issues were identified although some of these lines did not seem to match with any particular issues. This may well be a formatting problem that we are unable to see but thank the reviewer for pointing them out.

  1. Is the word ‘both’ redundant in line 73?

Response: Amended as suggested.

  1. Line 103.. did the authors mean thrombosis instead of haemostatic plug? Haemostatic plugs are formed mainly in the extravascular space while thrombosis is predominantly intravascular.

Response: We have changed this and thank the reviewer for this suggestion.

  1. Lines 195-197; authors should explain in one or 2 lines how adhesion contributes to prothrombotic environment.

Response: A short addition has been made to add some detail as requested.

  1. In concluding the review, what platelet mediators will the authors suggest targeting to control platelets’ contribution to thrombo-inflammation in DM

Response: We feel the addition of specific mediators of thrombo-inflammation may be beyond the scope of this review and have endeavoured to keep the conclusion concise, however reference the potential for investigating the pathways involved in mitochondrial dysfunction as a target and also developing miRNA/mRNA as clinical biomarkers.

Reviewer 2 Report

This paper “ Non-traditional pathways for platelet pathophysiology in diabetes: Implications for future therapeutic targets” is a very nice,  important and timely review by authors who are experimental experts in this area.

This referee has only a few questions which the authors are asked to address.

  • As presented in the review, the possible pathophysiological role of hyperactive platelets in diabetes ( primarily T2DM) has been studied for many years  both experimentally and clinically.  A major  question remains whether T2DM causes hyperactivation of platelets or independently hyperactive platelets worsen T2DM.  Is the effect of hyperactivated platelets reversed, when T2DM is metabolically normalized ?
  • How does insulin affect human platelets ? Directly via platelet IR or indirectly via other cells?
  • It is important, to precisely quantify platelet hyperactivation in various diseases. What are good, validated and robust methods to do this in both experimental and clinical studies ?

Author Response

We would like to thank the reviewer for the positive comments and have responded to the queries as below:

  • As presented in the review, the possible pathophysiological role of hyperactive platelets in diabetes (primarily T2DM) has been studied for many years  both experimentally and clinically.  A major  question remains whether T2DM causes hyperactivation of platelets or independently hyperactive platelets worsen T2DM.  Is the effect of hyperactivated platelets reversed, when T2DM is metabolically normalized ?

  1. The current review is focused on addressing the inflammatory role of platelets, in addition to the known thrombotic contribution to vascular pathology. The reviewer is raising an interesting point in relation to the effect of platelets on diabetes but this perhaps beyond the scope of the current review and requires a separate piece of work.
  2. The effects of improving glycaemia certainly reverses some of the abnormalities in platelet function, although not necessarily fully. This discussed in various parts of the review and we added a sentence in the conclusion to further clarify: “While some studies have shown reduced platelet activation by improving glycaemia, this appears to be partial with the added complication that aggressive glycaemic control induces hypoglycaemia, which is itself both prothrombotic and proinflammatory”.

  • How does insulin affect human platelets ? Directly via platelet IR or indirectly via other cells?
  1. Platelets can certainly be affected directly by insulin, given that that display insulin receptors, and this is discussed in section 3.1 in the current version of the MS. However, it is possible that insulin has indirect effects on platelets through modulation of the inflammatory milieu and this particular area requires further research.

  • It is important, to precisely quantify platelet hyperactivation in various diseases. What are good, validated and robust methods to do this in both experimental and clinical studies ?
  1. Quantifying platelet reactivity has been the focus of many studies and we are yet to adopt a reliable method that guides treatment decisions. While the point of the reviewer is well taken, this review is more focused on the inflammatory role of platelet, for which no tests are currently available. We added the following sentence: “A number of methods have been used to test the thrombotic properties of platelets, reviewed elsewhere, but tests to measure the inflammatory characteristics of these cells remain an area for future work.”   

Reviewer 3 Report

In this review Sagar and colleagues provide an interesting and timely discussion on the role of platelet metabolic reprogramming in enhancing the proinflammatory properties of these cells in people with diabetes.  This in turn may play a role in exacerbating the enhanced cardiovascular risk seen in this patient group.  I think the review highlights cutting edge ideas in the field and would be very beneficial to platelet and diabetes researchers.  The section on platelet metabolic reprogramming is particular interesting. Prior to publication, the review does need some proof-reading and revision just to improve the readability of the text – as there are many minor grammatical errors and sentences with excessive number of clauses that make some sections hard to read.  I apologise for the long list of minor comments here, but as I read through, I tried to make a list of simple changes that can be made to enhance the readability of this review.  I don’t believe any will need significant work and I hope could be sorted rapidly by the authors – so I am recommending minor revisions at this point. Thank you for a really interesting read.

Minor Comments

This is not essential, but I would suggest the authors consider moving Figure 2 up to the start of subsection 3 (just after the introductory text and prior to the start of 3.1) rather than having it at the end, as I think that would be a useful guide for readers to identify what is being described in this section.

Page (P) 1, Line (L) 14 -  Space between full stop and “This narrative..”

P1, L26 – missing ‘a’ – [a] first vascular event

P1, L29-31-   This line is missing some description “In the context of current evidence, the premature and more extensive vascular disease coupled with a prothrombotic environment in which platelet hyperactivity is thought to play a  key role” [in the reduced patient outcomes]?

P1, L34- Sentence unclearly stated, needs rewording  “The pervading view of the role of platelets is their  contribution to arterial thrombosis at sites of vascular rupture, with their activity targeted 35 by routine anti-platelet agents including aspirin, ticagrelor and clopidogrel [7-9].”  E.g. The pervading view of the role of platelets [in these worsened outcomes] is [due to] their contribution to arterial thrombosis at sites of [plaque] rupture[. These functions are routinely targeted by] anti-platelet agents…

P2, L40-51 – No references cited.  Needs some citations here.

P2 L51 – Platelet[s]-driven inflammation

P2, L55 – ‘..there is [a] coordinated response….’

P2, L59 – delete ‘,beyond their thrombotic effects,’

P2, L65 delete ‘gaining an understanding’ and replace ‘identify’ to ‘identifying’

P2, L76-78 – change ‘The tonic release of these mediators acts to ensure platelet quiescence  and prevent platelet-mediated immune cell infiltration of the subendothelial space, factors that are critical to preventing vascular inflammation ‘ to ‘The tonic release of these mediators prevents vascular inflammation by ensuring platelet quiescence and preventing platelet-mediated cell infiltration of the subendothelial space’   

P2, L81 – Replace ‘In disease, endothelial cells express and number of cell adhesion molecular and chemotactic messengers that…’ with ‘When inflamed, endothelial cells increase the cell surface expression of cell adhesion molecules and release of chemotactic messengers that…’

P2, L89 – Replace ‘increasing the highly inflammation’ with ‘enhancing the proinflammatory’ (or equivalent)

P2-3, L90-91 – delete ‘amongst other mechanisms’ and amend to ‘to drive clot formation’

P3, L91-93 – This statement on endothelial dysfunction should follow the previous statement on endothelial dysfunction on L81.

P3, L100 – “an [increased] potential for…”

P3, L101 – change ‘contributes’ to ‘contributing’

P3, L121 – delete ‘up to’

P4, L126-127 – Replace ‘intra-platelet cyclic adenosine monophosphate (cAMP)’ with ‘cytosolic cyclic

adenosine monophosphate (cAMP) concentration’

P4, L127 – Replace ‘The increased cAMP’ with ‘The increased cytosolic cAMP concentration’

P4, L130 – replace ‘unopposed’ with ‘disinhibited’ – there are other pathways other than insulin that keep platelet function in check.

P4, L134 – Replace ‘intra-platelet’ with ‘cytosolic’.  Cytosolic is more accurate than intra-platelet, which could imply changes in cytosolic or organellar calcium concentration.

P4, L137-164. This is difficult to read, and reads more like a list of all the ways that NO bioavailability is decreased and the consequences of that.  I feel this could be written more simply, and may benefit with a summary diagram to complement it.

P4, L166 – delete ‘,beyond NF-KB,’

P4, L173-175.  Elevated PKC [activity] has been demonstrated in the platelets taken from healthy volunteers [left] in hyperglycaemic conditions.’

P5, L188 – Replace ‘compared with healthy controls’ with ‘compared to platelets taken from healthy people’  -  ‘healthy controls’ is a very dehumanising way to describe our donors.

P5, L199 – delete ‘In respect to DM,’

P7, L288-290.  “Given the relative similarities between the metabolic phenotype of both immune cells and platelets, it can be hypothesised that similar immune changes occur in these two cell types.” – I feel that this statement needs further clarification.  There is no evidence or citation provided to support the assertion that immune cells and platelets have a similar metabolic phenotype.  I am also unclear what is meant by “immune changes”.  I think this needs rewriting with additional citations.

P7, L292 – ‘…is fundamentally metabolic and energy-requiring involving the generation of ATP..’ – This is awkwardly phrased.  This can be stated more simply along the lines of “Platelet activation in response to both thrombotic and inflammatory process is energetically expensive, requiring a significantly enhanced generation of ATP via…”

P7, L293-295 ‘Disease states have been shown to influence these key metabolic processes with elevated extracellular acidification rate (ECAR), demonstrating increased glycolysis and oxygen consumption rate (OCR), indicating oxidative phosphorylation’.  It would be better to be more specific about what disease states elicit these changes.  I think this can be worded better in something along the lines of “[Insert specific disease states] have been shown to increase platelet glycolysis and oxidative phosphorylation,  as assessed through demonstration of enhanced ECAR and OCR respectively”.  This might help the readers as I’m not sure everyone will be familiar with ECAR in particular.

P7, L297 ‘therefore hyperglycaemia in DM may drive[s] these processes, [whilst] and improv[ed]ing glycaemic control can reverse these changes’

P8, L311 – remove the unwanted space in ‘interes t’

P8, L314 – ‘with[in the] inflammatory….’

P8, L319 – ‘,also referred to as ‘immunometabolic reprogramming’  I would suggest removing this or using this term instead of what is current there, as I’m not sure you need both given they are essentially rewording of the same term.

P8, L323 – ‘have [an] altered platelet transcriptome’

P8, L324 – do you mean “enhanced” expression of ABCC4, rather than ‘excess’?

P8, L328-329 – ‘with platelets from [people with] DM’

P8, L337 – Although the gene is SELP, is it clearer to say P-selectin mRNA rather than selectin-P?

P9, L354 – SELP is the gene, P-selectin is the protein – so I’m not sure SELP is the appropriate term here.

P10, L412. The conclusion is quite long and the first paragraph seems to reiterate parts of the introduction.  I think this could be written more concisely to better identify the future prospects of this research area.

Author Response

The authors would like to thank the Reviewer for the encouraging and positive comments about the review. We are very grateful for the thorough and helpful suggestions and comments which have all been addressed and outlined below:

Page (P) 1, Line (L) 14 -  Space between full stop and “This narrative..”

Response: Amended accordingly

P1, L26 – missing ‘a’ – [a] first vascular event

Response: Amended accordingly

P1, L29-31-   This line is missing some description “In the context of current evidence, the premature and more extensive vascular disease coupled with a prothrombotic environment in which platelet hyperactivity is thought to play a  key role” [in the reduced patient outcomes]?

Response: Amended accordingly

P1, L34- Sentence unclearly stated, needs rewording  “The pervading view of the role of platelets is their  contribution to arterial thrombosis at sites of vascular rupture, with their activity targeted 35 by routine anti-platelet agents including aspirin, ticagrelor and clopidogrel [7-9].”  E.g. The pervading view of the role of platelets [in these worsened outcomes] is [due to] their contribution to arterial thrombosis at sites of [plaque] rupture[. These functions are routinely targeted by] anti-platelet agents…

Response: Amended accordingly and we thank the Reviewer for the helpful rewording of these sentences to provide clarity.

P2, L40-51 – No references cited.  Needs some citations here.

Response: Citations added and thanks to the author for highlighting these were missing.

P2 L51 – Platelet[s]-driven inflammation

Response: Amended accordingly

P2, L55 – ‘..there is [a] coordinated response….’

Response: Amended accordingly

P2, L59 – delete ‘,beyond their thrombotic effects,’

Response: Amended accordingly

P2, L65 delete ‘gaining an understanding’ and replace ‘identify’ to ‘identifying’

Response: Amended accordingly

P2, L76-78 – change ‘The tonic release of these mediators acts to ensure platelet quiescence  and prevent platelet-mediated immune cell infiltration of the subendothelial space, factors that are critical to preventing vascular inflammation ‘ to ‘The tonic release of these mediators prevents vascular inflammation by ensuring platelet quiescence and preventing platelet-mediated cell infiltration of the subendothelial space’  

Response: Amended accordingly, with thanks to the Reviewer for the improved wording.

P2, L81 – Replace ‘In disease, endothelial cells express and number of cell adhesion molecular and chemotactic messengers that…’ with ‘When inflamed, endothelial cells increase the cell surface expression of cell adhesion molecules and release of chemotactic messengers that…’

Response: Amended accordingly

P2, L89 – Replace ‘increasing the highly inflammation’ with ‘enhancing the proinflammatory’ (or equivalent)

Response: Amended accordingly

P2-3, L90-91 – delete ‘amongst other mechanisms’ and amend to ‘to drive clot formation’

Response: Amended accordingly

P3, L91-93 – This statement on endothelial dysfunction should follow the previous statement on endothelial dysfunction on L81.

Response: Moved to follow the above sentence as suggested by the Reviewer.

P3, L100 – “an [increased] potential for…”

Response: Amended accordingly

P3, L101 – change ‘contributes’ to ‘contributing’

Response: Amended accordingly

P3, L121 – delete ‘up to’

Response: Amended accordingly

P4, L126-127 – Replace ‘intra-platelet cyclic adenosine monophosphate (cAMP)’ with ‘cytosolic cyclic

adenosine monophosphate (cAMP) concentration’

Response: Amended accordingly

P4, L127 – Replace ‘The increased cAMP’ with ‘The increased cytosolic cAMP concentration’

Response: Amended accordingly

P4, L130 – replace ‘unopposed’ with ‘disinhibited’ – there are other pathways other than insulin that keep platelet function in check.

Response: Amended accordingly

P4, L134 – Replace ‘intra-platelet’ with ‘cytosolic’.  Cytosolic is more accurate than intra-platelet, which could imply changes in cytosolic or organellar calcium concentration.

Response: Amended accordingly and thanks to the Reviewer for adding accuracy to these statements.

P4, L137-164. This is difficult to read, and reads more like a list of all the ways that NO bioavailability is decreased and the consequences of that.  I feel this could be written more simply, and may benefit with a summary diagram to complement it.

Response: These sentences have been simplified to add clarity. This section is incorporated into figure 1 and an additional figure may take focus away from more specific areas of the review. However, we will include this extra figure if the reviewer feels it is essential.

P4, L166 – delete ‘,beyond NF-KB,’

Response: Amended accordingly

P4, L173-175.  Elevated PKC [activity] has been demonstrated in the platelets taken from healthy volunteers [left] in hyperglycaemic conditions.’

Response: Amended accordingly

P5, L188 – Replace ‘compared with healthy controls’ with ‘compared to platelets taken from healthy people’  -  ‘healthy controls’ is a very dehumanising way to describe our donors.

Response: Amended accordingly.

P5, L199 – delete ‘In respect to DM,’

Response: Amended accordingly

P7, L288-290.  “Given the relative similarities between the metabolic phenotype of both immune cells and platelets, it can be hypothesised that similar immune changes occur in these two cell types.” – I feel that this statement needs further clarification.  There is no evidence or citation provided to support the assertion that immune cells and platelets have a similar metabolic phenotype.  I am also unclear what is meant by “immune changes”.  I think this needs rewriting with additional citations.

Response: These sentences have been re-worded for clarity and relevant citations now added

P7, L292 – ‘…is fundamentally metabolic and energy-requiring involving the generation of ATP..’ – This is awkwardly phrased.  This can be stated more simply along the lines of “Platelet activation in response to both thrombotic and inflammatory process is energetically expensive, requiring a significantly enhanced generation of ATP via…”

Response: Amended accordingly

P7, L293-295 ‘Disease states have been shown to influence these key metabolic processes with elevated extracellular acidification rate (ECAR), demonstrating increased glycolysis and oxygen consumption rate (OCR), indicating oxidative phosphorylation’.  It would be better to be more specific about what disease states elicit these changes.  I think this can be worded better in something along the lines of “[Insert specific disease states] have been shown to increase platelet glycolysis and oxidative phosphorylation,  as assessed through demonstration of enhanced ECAR and OCR respectively”.  This might help the readers as I’m not sure everyone will be familiar with ECAR in particular.

Response: This sentence has been re-worded/re-structured in line with the Reviewer’s useful comments.

P7, L297 ‘therefore hyperglycaemia in DM may drive[s] these processes, [whilst] andimprov[ed]ing glycaemic control can reverse these changes’

Response: Amended accordingly

P8, L311 – remove the unwanted space in ‘interes t’

Response: Amended accordingly

P8, L314 – ‘with[in the] inflammatory….’

Response: Amended accordingly

P8, L319 – ‘,also referred to as ‘immunometabolic reprogramming’  I would suggest removing this or using this term instead of what is current there, as I’m not sure you need both given they are essentially rewording of the same term.

Response: Amended accordingly

P8, L323 – ‘have [an] altered platelet transcriptome’

Response: Amended accordingly

P8, L324 – do you mean “enhanced” expression of ABCC4, rather than ‘excess’?

Response: The reviewer is correct and this has been amended accordingly

P8, L328-329 – ‘with platelets from [people with] DM’

Response: Amended accordingly

P8, L337 – Although the gene is SELP, is it clearer to say P-selectin mRNA rather than selectin-P?

Response: Amended accordingly

P9, L354 – SELP is the gene, P-selectin is the protein – so I’m not sure SELP is the appropriate term here.

Response: Amended accordingly

P10, L412. The conclusion is quite long and the first paragraph seems to reiterate parts of the introduction.  I think this could be written more concisely to better identify the future prospects of this research area.

Response: We have shortened the initial part of the conclusion and re-worded sentences to make the conclusion more concise.